# First Trifluoromethylated Phenanthrolinediamides: Synthesis, Structure, Stereodynamics and Complexation with Ln(III)

**DOI:** 10.3390/molecules27103114

**Published:** 2022-05-12

**Authors:** Yuri A. Ustynyuk, Pavel S. Lemport, Vitaly A. Roznyatovsky, Konstantin A. Lyssenko, Alexey O. Gudovannyy, Petr I. Matveev, Ennie K. Khult, Mariia V. Evsiunina, Vladimir G. Petrov, Igor P. Gloriozov, Anton S. Pozdeev, Valentine S. Petrov, Nane A. Avagyan, Alexander S. Aldoshin, Stepan N. Kalmykov, Valentine G. Nenajdenko

**Affiliations:** 1Department of Chemistry, Moscow State University, 1-3 Leninskiye Gory, GSP-1, 119991 Moscow, Russia; yuriustynyuk@gmail.com (Y.A.U.); lemport.pavel@yandex.ru (P.S.L.); vit.rozn@nmr.chem.msu.su (V.A.R.); klyssenko@gmail.com (K.A.L.); alexeygudovannyy@gmail.com (A.O.G.); petr.i.matveev@gmail.com (P.I.M.); mashko-ya-e@mail.ru (M.V.E.); vladimir.g.petrov@gmail.com (V.G.P.); iiglor@nmr.chem.msu.ru (I.P.G.); anton.pozdeev.1998@gmail.com (A.S.P.); vs.petrov25@gmail.com (V.S.P.); nane.avakyan@mail.ru (N.A.A.); aldon2258@mail.ru (A.S.A.); stepan_5@hotmail.com (S.N.K.); 2Department of Materials Science, Moscow State University, 1-73 Leninskiye Gory, GSP-1, 119991 Moscow, Russia; jennie.hult@gmail.com

**Keywords:** phenanthroline, fluorine-containing, N,O-hybrid ligands, X-ray diffraction, NMR, DFT calculations, stereodynamics, complexes, lanthanoids

## Abstract

The first examples of 1,10-phenanthroline-2,9-diamides bearing CF_3_-groups on the side amide substituents were synthesized. Due to stereoisomerism and amide rotation, such complexes have complicated behavior in solutions. Using advanced NMR techniques and X-ray analysis, their structures were completely elucidated. The possibility of the formation of complex compounds with lanthanoids nitrates was shown, and the constants of their stability are quantified. The results obtained are explained in terms of quantum-chemical calculations.

## 1. Introduction

Amides of 1,10-phenanthroline-2,9-dicarboxylic acid are an important type of tetradentate ligands widely used in modern coordination chemistry. Ligands of this type are capable of forming strong complexes with large cations (ionic radius ≈ 1 Å) in highly acidic media due to the combination of soft nitrogen centers and hard oxygen centers (within the framework of the HSAB concept) in the coordination node. These ligands have large size of the coordination cavity and relatively low Brønsted basicity [1]. Varying the structure of amide fragments and the substituents in the heterocyclic core enables “fine tuning” of the ligand to the requirements of the guest cation. As a result, such an approach is very fruitful in the design of efficient catalytic and solvent extraction systems. In particular, highly selective phenanthroline extractants have been developed for the separation of lanthanoids and minor actinides (Am, Np, Cm) [2,3,4,5], which is one of the most urgent and difficult problems in the development of a closed fuel cycle in nuclear power industry. The high coordinating ability of phenanthroline derivatives towards transition metals made them attractive platforms on which the coordinated metal can serve as a Lewis acid binding site for various substrates. That is why functionally substituted phenanthrolines are increasingly used in asymmetric catalysis (see reviews [6,7,8,9,10,11,12,13]). In recent years, a large number of lanthanoid and rare-earth elements complexes with chiral donor ligands have been obtained [13,14,15]. For example, a scandium complex of C_2_-symmetric phenanthroline derived diol has been successfully used in the enantioselective ring opening of epoxides and amination of β-ketoesters [16]. Recently, the synthesis of some representatives of phenanthrolines containing a fluorine atom in the aromatic core was reported [17]. However, chiral phenanthroline diamides have not yet been described. The use of such ligands in catalysis could open new interesting prospects.

Continuing our studies of phenanthroline diamides [18,19,20,21,22,23,24], in this work, we have carried out a synthesis of 2-trifluoromethylpyrrolidine derived diamides **1** and **2**. Both compounds were prepared by acylation of 2-CF_3_-pyrrolidine by phenanthroline derived acyl chlorides. These ligands contain two chiral centers of the same type in the amine fragments and therefore they exist as a mixture of diastereomers (meso-form (*R,S*) and racemate (mixture of *R,R*- and *S,S*-enantiomers)) (Figure 1). This study contains the results of NMR, X-ray and computational study of these new ligands as well as their complexation with La(III), Eu(III), Nd(III) and Lu(III).

It should be noted that the application of fluorinated compounds is an important trend of modern chemistry and materials science. In particular, the incorporation of fluorine into the target molecule is recognized as an effective tool in the development of new drugs and materials science. Many important characteristics, such as lipophilicity, metabolism, membrane permeability, binding efficiency and bioavailability can be altered by the inclusion of fluorine. Fluorinated materials have much higher resistance to oxidation, light degradation and hydrolysis. The unification of fluorine chemistry and heterocycle chemistry is especially fruitful for opening new horizons in the synthetic organic chemistry [25,26,27]. Another important advantage providing by the presence of fluorine in a molecule is possibility to use ^19^F NMR. Nowadays, this method became a very powerful tool for various structural studies including complex natural objects. This is due to such advantages of ^19^F NMR as 100% abundance of ^19^F isotope, very broad chemical shifts span (800 ppm) and high sensitivity of this method. Incorporation of CF_3_-groups significantly simplified the study of the stereodynamics of these compounds using ^19^F NMR.

## 2. Results and Discussions

### 2.1. Synthesis, Structure and Stereodynamics of Ligands 1 and 2

Synthesis of the “parent” pyrrolidine-derived ligand was first published in 2004 [28]. Later, some properties of this compound were revealed [29].

Starting from (±)-2-(trifluoromethyl)pyrrolidine [30], we obtained new phenanthroline ligands **1** and **2**. Then we started their structural study. Both compounds were characterized by combination of spectral methods and high-resolution mass spectrometry. They are white powders, readily soluble in dichloromethane, chloroform, acetone and moderately soluble in acetonitrile and hexane. In spite of simple structure of prepared diamides **1** and **2**, their NMR spectra were surprisingly complicated in CDCl_3_ solution at room temperature (Appendix A) even taking into account that mixtures of diastereomers are under study. The ^19^F-NMR spectrum of **1** (Figure 2a) contains a set of 8 signals of CF_3_ groups the doublet splitting of which is due to the *^3^J_19F,1H_* spin-spin coupling. Simpler picture is observed in the spectrum of the heteronuclear double resonance ^19^F-{^1^H} (Figure 2b). Eight singlets of CF_3_ groups of different intensities are clearly seen in it. Eight signals of CO-groups are also present in the carbonyl region of the ^13^C-NMR spectrum of **1** (Figure 2c).

Such complex ^19^F and ^13^C-NMR spectra indicate that the internal rotation along Phen-CO bonds in diamides **1** and **2** in solutions at 23 °C occurs as a fast process in the NMR time scale, while the rotation along the amide bonds N–C=O is completely inhibited. Thus, diamides **1** and **2** in this respect behave similarly to other ligands of this type that we studied earlier [23]. As a result, three rotamers along the amide bonds (**A**–**C**) (Figure 3), differing in the orientation of -CHCF_3_ fragments relative to the phenanthroline backbone, which coexist in solutions, give separate signals in the NMR spectra. Note that processes of this type in arylcarboxamides are well documented [31,32,33,34]. In contrast to symmetric structures **A** and **C**, the rotamer **B** contains two non-equivalent CF_3_-groups in the structure. This consideration allows us to explain very well the observed complicated spectra.

To have deeper insight in the equilibrium between rotamers of ligand **1**, we decided to separate diastereomers in pure form. The quantitative separation of diastereomers of ligand **1** was carried out by HPLC in the acetonitrile/water system. The chromatogram contains two closely spaced peaks with an intensity ratio of 2:3 (see Appendix A). At slow isothermal concentration of the acetonitrile solution of the second fraction, crystals of the racemate of **1** suitable for X-ray were obtained (Figure 4). Depending upon the crystal growth procedure, we were able to investigate two racemic crystal forms of ligand **1,** which corresponds to isomer **1A** and the hydrate of isomer **1B** with one water molecule.

The analysis of these two forms clearly shows that they are differ by orientation of pyrrolidine ring in respect to phenanthroline. In **1A** and **1B**·H_2_O the C=O groups are in trans orientation in respect to nitrogen atoms of phenanthroline while the CF_3_–C–N–C=O torsion angles in **1A** and **1B**·H_2_O differ. In **1** CF_3_–C groups in both pyrrolidine substituents are in syn-periplanar orientation in respect to C=O groups while in **1B**·H_2_O one pyrrolidine group is syn-periplanar while the other is in the antiperiplanar orientation. The variation of mutual orientation of the trifluoromethylated pyrrolidine almost does not affect the bond lengths distribution in amide fragment (see Table 1) but leads to different weak interactions between the trifluoromethylated pyrrolidine and nitrogen of phenanthroline ring. As one can see in the case of **1A** N(1) and N(10) atoms participate in formation of the weak contacts with CH_2_ group of pyrrolidine ring while in **1B**·H_2_O one of C–H…N contacts is formed by more acidic C-HCF_3_ group. Basing on the geometric parameters one can propose that latter contact with H…N distance equal 2.16 Å should be stronger than those (2.36 Å) in **1A** and thus such conformation should be more stable.

At the same time, we cannot exclude that stabilization of the conformation in **1B**·H_2_O is the consequence of crystal packing effects (Figure 5). Analysis of crystal packing have revealed that both molecules participate in the formation of infinite stacks with comparable interplane separation (3.38 vs. 3.40 Å) but slightly different area of overlap.

Furthermore, the infinite stacks in hydrate are additionally interlinked by O–H…O hydrogen bonds (O…O 2.840(4)-2.859(4) with water molecule which are absent in **1A**.

In order to estimate the relative stability of the conformations of the molecule in **1A** and **1B**·H_2_O we performed DFT (PBE/def-2-TZVP) calculations. The geometry of **1A** was optimized using the very tight optimization criteria and empirical dispersion corrections on the total energy [35] with the Becke-Johnson damping (D3) [36]. The optimization of **1A** lead to the geometry that is almost identical to those in **1B**·H_2_O (see Table 1). Thus, we can assume that this geometry corresponds to global minimum. In order to estimate the possible way of conformation transformation from **1B**·H_2_O to the conformation in **1A** we check two possible opportunities and performed the relaxed potential energy scan along the Phen–C=O and O=C–N–C–CF_3_ torsion angles (step equal to 10°). The barrier to rotation for the first coordinate is equal to 14 kcal/mol while for the other it can be as much as 24 kcal/mol. Upon the relaxed scan we have found the additional minimum which geometry is almost identical to those observed in **1A** (see Table 1). The difference in energy of these two conformers is only 1.04 kcal/mol.

Aiming at estimating the energy of C–H…N contacts in two conformers we have used the topological analysis of the electron density distribution function ρ(**r**) within Bader’s quantum theory of “Atoms in Molecule’’ (QTAIM) theory [37]. Using the AIM formalism, one can distinguish the binding interatomic interactions from all other contacts. When the distribution of ρ(**r**) in molecule or crystal is known, one can answer the question whether the bonding interaction is present or not by the search of the bond critical point (3,–1) and predict the energy of weak intermolecular interactions (Econt) with high accuracy on the basis of the potential energy density function v(**r**) – the correlation suggested by Espinosa et al. (CEML) [38]. Recently, the physical interpretation of CEML was suggested [39].

According to the critical point (CP) search of ρ(**r**), CP (3,–1) in conformers in **1A** and **1B**·H_2_O are located not only for all expected bonds but also for weak C–H…N interactions and for series of F…H and H…H interactions between the trifluoromethylated pyrrolidine moieties (Figure 6). Expectedly, all C–C, C–N, C–H, C–O and C–F bonds are characterized by the negative values of ∇^2^ρ(**r**) and the negative electron energy density (h_e_(**r**)) in CP(3,–1) and, therefore, correspond to shared type of interatomic interactions. In contrast all weak H…H, N…H and H…F interactions are characterized by both positive ∇^2^ρ(**r**) and h_e_(**r**) in CP (3,–1) and thus correspond to the closed-shell interactions. The energy of H…H and H…F interaction in both conformers is comparable and vary in the range of 0.4–0.9 kcal/mol. In their turn, the energy of C–H…N contacts is higher and equal to ca. 3.2 and 4.6 kcal/mol for CH_2_ and CF_3_CH group. As one can see, the difference in energy of this interactions is very close to the estimated difference of the conformers’ energy.

Thus, the stabilization of conformation is mainly governed the intramolecular H-bond and, although the difference in energy of two conformers is negligible, the conformation obtained in **1B**·H_2_O was also observed in the case of meso-**2B,** which crystallized without any solvate molecules (Figure 7).

The interesting feature of **2B** is that, due to presence of chlorine substituent, the formation of stacking interaction (with interplane distance ca. 3.3 Å) is accompanied by the Cl…π interaction (Figure 8).

As expected, the ^19^F NMR spectrum of racemic **1** (Figure 9) is simpler, and consists of 4 doublets in the region from –73.9 ppm up to –74.8 ppm. The spectrum of rotamer **B** should contain two signals of nonequivalent CF_3_ groups of equal intensity, while rotamers **A** and **C** should have in their ^19^F NMR spectra by one doublet each since both CF_3_ groups of these rotamers are equivalent. These considerations made it possible to distinguish signals from rotamer **B** but do not allow the two remaining doublets to be assigned (Figure 9).

The assignment of signals of two different CF_3_-groups of rotamer **B** (**B1** and **B2**) in the spectrum of racemate was unambiguously confirmed by EXSY ^19^F NMR (Figure 10).

The slow exchange of the positions of CF_3_-groups is observed due to hindered internal rotation along the amide bonds in EXSY spectrum. As a result, the spectrum contains off-diagonal peaks between signals linked to each other by one exchange act. Such peaks are present only between the **A→C, B1, B2→C** doublets since only one rotation around the amide bond is required for the transitions between these rotamers. The EXSY spectrum at short mixing times has no **B1→B2** cross-peaks because two acts of exchange are required for such transition. Consequently, peaks **B1** and **B2** belong to rotamer **B** which is realized in the crystal (Figure 6).

Next, the influence of temperature on the ligand **1** rotamers equilibrium was studied in toluene-d_8_. Rising of temperature results in increase of content of rotamers **A** and **C** (Table 2) in the equilibrium mixture.

A significant increase in the rate of rotation around the amide bonds is clearly observed above 40 °C. As a result, it is possible to observe in the spectra a gradual broadening and merging of signals from rotamers of each of the diastereomers with increasing of temperature, and two very broad signals are seen at 75 °C (Figure 11).

The potential energy surfaces of rotamers **A**–**C** of both diastereomers are rather complex. There are several very close local minima differing in dihedral angles for Phen–CO bonds (the difference in free energies is ±0.5 kcal/mol) near the global minima. The structures corresponding to the global minima are shown in Figure 12.

As it can be seen, all rotamers **A–C** have practically similar stability in the gas phase, but they differ significantly in their dipole moments. Therefore, it is possible to expect that the equilibrium between rotamers can be shifted by solvent polarity change. Indeed, measurement of spectra in four different solvents (toluene, chloroform, acetone and nitrobenzene) confirmed our proposal. Rising the polarity of the solvent leads to an increase in the content of the most polar rotamers **A** and **B**, while the content of non-polar rotamer **C** decreases (Table 3).

Based on these data, one can assign the signals of rotamers **A** and **C**. As a result, we have clear understanding of spectral data for mixture of rotamers **A–C**. The results obtained for a solution in toluene fall out somewhat from the general dependence. This means that, although the polarity of the solvent is the main factor determining the equilibrium position, other factors, such as the solvent ability to form hydrogen bonds, should be also considered.

We were unable to obtain crystals of the meso-form of **1** suitable for X-ray studies. However, having solved the problem of assigning the signals of the rotamers of the racemic **1**, it was possible to make the unambiguous assignment of the signals of rotamers of the meso-form as well. The ^19^F–{^1^H}–NMR spectrum in CDCl_3_ is given in Figure 2b. Two singlets of equal intensity at –73.75 ppm and –74.81 ppm are related to rotamer **B** (72% at 21 °C). Two singlets at –73.73 ppm and –74.19 ppm belong to rotamers **A** and **C** (14% each at 21 °C). The assignment of signals in this spectrum was done according to two-dimensional NMR spectra (see Appendix A).

Having established the approach for structure elucidation applied for ligand **1**, we were able to determine the structure and composition of diastereoisomers of diamide **2** using a combination of ^1^H, ^13^C and ^19^F NMR spectra (Table 4).

The ^19^F NMR spectra of mixture of diastereomers of diamides **1** and **2** are substantially similar (Figure 2 and Figure 13). As a result, it possible to distinguish all groups of lines in the spectrum of diamide **2** related to rotamers **A–C** for both the meso-form and the racemate. All 8 singlets of CF_3_-groups belonging to 6 rotamers are present in the ^19^F–{^1^H}–NMR spectrum in CDCl_3_ (Figure 13). The ratio of two diastereomers of compound **2** is 2:3 according to the integration data. Thus, ligands **1** and **2** have the same ratio of diastereomers.

### 2.2. Complexation of Ligands **1** and **2** with Ln(III) Nitrates

The prepared amides **1** and **2** are highly attractive ligands for various cations. Based on ligand **1**, we obtained a series of complexes with nitrates of La, Nd, Eu, and Lu. It was expected that such interaction can lead to complexes of both 1:1 and 2:1 **L**:Ln stoichiometry. Using acetonitrile as a solvent for this reaction we were able to synthesize **L**·Ln(NO_3_)_3_ complexes. Complexes of diamide **1** with nitrates La (III), Nd (III), Eu (III), and Lu (III) of 1:1 composition were isolated from solutions in acetonitrile in solid form as light-colored powders. We studied all these complexes using IR and ^19^F NMR spectroscopy. The coordination of the metal leads to a bathochromic shift of the ν_C=O_ stretching vibration band in the IR spectrum by 42 cm^−1^ in the lanthanum complex. This shift rises with an increase in the atomic number of the cation to reach 49 cm^−1^ for Lu complex. Some characteristics of **1**·Ln(NO_3_)_3_ complexes are given in Table 5.

^19^F-NMR spectra of **1**·Ln(NO_3_)_3_ complexes have very much in common (Figure 14). Due to formation of coordination bonds, the amide rotation stops in these cases. This results in two doublet signals are observed in each spectrum which belongs to a complex of *rac*- and *meso*- forms, respectively.

Having confirmed the possibility of the formation of complexes for the ligands obtained we measured the stability constants by the UV–Vis spectrophotometric titration method. The stability constants were determined by spectrophotometric titration in the UV–visible region (Table 6, Figure 15).

Figure 15 represents an example of spectrophotometric data of 1 and Eu(NO_3_)_3_ complexation in acetonitrile (see Appendix A for analogues data for all the titrations). More basic diamide 1 forms complexes of stoichiometry 1:1 and 2:1 with cations of early lanthanoids La(III) and Nd(III) with larger ionic radius. The complexation with lanthanoids of the middle (Eu) and the end of the series (Lu) results in only 1:1 complexes. Less basic diamide 2 contains two electron-withdrawing chlorine atoms at positions 4 and 7 of the phenanthroline nucleus. As a result, this ligand forms only 1:1 complexes with all studied lanthanoids. The stability constants for both ligands increase with an increase in the atomic number of the cation. As expected, the log β_1_ for diamide 2 complexes are an order of magnitude lower than the stability constants of diamide 1 complexes.

Additionally, we obtained electrostatic potential (ESP) maps at B3LYP/G-31G(d,p) theoretical level (Gaussian 16 [40]) for both ligands. ESP maps in two projections are given in Figure 16 (a single potential scale from −0.01 to +0.015 conventional units). The introduction of chlorine in a phenanthroline system leads to a significant change in electron density of the molecules.

Merz-Kollman (ESP) charges of some atoms for optimized geometries of 1 and 2 are shown in Table 7.

Thus, less effective complexation of **2** with lanthanoids can be associated with the decreased charges at amide oxygens and phenanthroline nitrogens of **2** in comparison to diamide **1**, because it leads to weaker ionic component to Ln–O and Ln–N bonds.

## 3. Materials and Methods

### 3.1. Materials

Ar-saturated solvents, purified and dried using standard techniques were used [41]. Lanthanoids nitrates La(NO_3_)_3∙_6H_2_O, Nd(NO_3_)_3∙_6H_2_O, Eu(NO_3_)_3∙_6H_2_O and Lu(NO_3_)_3∙_xH_2_O were purchased from Sigma-Aldrich, Co. (St. Louis, MO, USA) and used without further purification. Water content x in lutetium nitrate was determined as x = 3. Deuterated solvents for NMR spectra were purchased from Cambridge Isotope Laboratories, Inc. (Andover, MA, USA) and used without further purification. Triethylamine was purified by simple distillation, previously held for 12 h over sodium hydroxide. (±)-2-(Trifluoromethyl)pyrrolidine was synthesized according to known method [30]. 1,10-Phenanthroline-2,9-dicarbonyl chloride and 4,7-dichloro-1,10-phenanthroline-2,9-dicarbonyl chloride for synthesis of ligands **1** and **2**, correspondingly, may be prepared according to known procedures [23].

### 3.2. Methods

NMR spectra were recorded at room temperatures (if otherwise not stated) using standard 5 mm sample tubes on Agilent 400–MR spectrometer equipped with OneNMR and ATB probes with operating frequencies of 400.1 MHz (^1^H), 100.6 MHz (^13^C) and 376.0 MHz (^19^F). The concentrations of ligands and their complexes were 20 g·L^−1^. For NMR of racemic **1** the concentration was 1 g·L^−1^. Deuterated solvents for NMR spectra were purchased from commercial sources and used without further purification.

IR spectra were recorded on FTIR spectrometer Nicolet iS5 (Thermo Scientific) using an internal reflectance attachment with diamond optical element – attenuated total reflection (ATR) with 45° angle of incidence. Resolution 4 cm^−1^, the number of scans is 32.

HRMS ESI (+) mass spectra were recorded on the MicroTof Bruker Daltonics and Orbitrap Elite instruments.

Single crystals of **1A**, **1B·**H_2_O and **2B** were obtained upon slow isothermal (25 °C) recrystallization from MeCN.

The crystallographic data was collected using Bruker Quest D8 diffractometer equipped with a Photon-III area-detector (shutterless ϕ- and ω-scan technique), using Mo K_a_-radiation. The intensity data were integrated by the SAINT program and corrected for absorption and decay by SADABS. Structures were solved by direct methods using SHELXT and refined against F^2^ using SHELXL-2018.

Detailed crystallographic data provided here in the Appendix A.

UV−Vis spectra were recorded at the temperature 25.0 ± 0.1 °C in the wavelength region of 200−400 nm (0.5 nm interval) on a Shimadzu UV 1800 spectrophotometer controlled by LabSolutions UV−Vis software with thermostatic attachment (Shimadzu TCC-100) using quartz cuvettes with an optical path length of 10 mm. A stock solution of the ligand was prepared (ca. 10^−4^ mol/L) by dissolving respective ligand in CH_3_CN, and then a working ligand solution (ca. 10^−5^ mol/L) was prepared from the initial solution. A working titrant solution (10^−3^ mol/L) was prepared by dissolving corresponding lanthanoid (III) nitrate hydrate Ln(NO_3_)_3_·6H_2_O (Ln = La, Nd, Eu, Lu) in CH_3_CN. Acetonitrile (CH_3_CN; 99.95%, HPLC grade, Panreac AppliChem) was dried over molecular sieves (zeolite KA, 3 Å, balls, diameter 1.6−2.5 mm, production HKC Corp., Hong Kong) prior to use. The titration was carried out by adding 2 μL aliquots of the working metal cation solution to 2 mL of the working ligand solution in the titration cell. The titration continued until no obvious change was observed in the spectra. The stability constants of the Ln(III) complexes were calculated using the HypSpec2014 program.

### 3.3. Synthesis and Analytical Data

#### 3.3.1. Synthesis of Diamides

A solution of (±)-2-(trifluoromethyl)pyrrolidine (12.5 mmol) and 1.74 mL of triethylamine (12.5 mmol) in 10 mL of methylene chloride was added at –10 °C under vigorous stirring to a suspension of 5 mmol of the corresponding chloride in 50 mL of methylene chloride. Then the reaction mixture was allowed to reach room temperature followed by refluxing for 4 h. Next, the reaction mixture was diluted with 50 mL of methylene chloride, washed with water (2 × 50 mL), dried over sodium sulfate, and the solvent was distilled off. The residue was purified by recrystallization from a mixture of hexane/ethyl acetate or by silica gel column chromatography (eluent: hexane/acetone = 7/3), obtaining the desired product in the form of a white or slightly colored solid.

*(2-(trifluoromethyl)pyrrolidin-1-yl)(9-((2-(trifluoromethyl)pyrrolidin-1-yl)carbonyl)-1,10-phenanthrolin-2-yl)methanone* (**1**). Yield = 75% (1.91 g), white powder, m.p. 190–195 °C, R_f_ 0.35 (hexane:acetone 2:1); ^1^H NMR (400 MHz, CDCl_3_) δ 8.46–8.32 (m, 2H, phenanthroline CH), 8.32–8.11 (m, 2H, phenanthroline CH), 7.92–7.77 (m, 2H, phenanthroline CH), 7.22–5.11 (group of m, 2H, pyrrolidine ring CH), 4.85–3.00 (group of m, 4H, pyrrolidine ring CH_2_), 2.62–1.87 (group of m, 8H, pyrrolidine ring CH_2_); ^13^C NMR (101 MHz, CDCl_3_) δ 168.2–165.5 (group of CO), 152.8–151.9, 143.9–143.2, 137.3–136.4, 129.3–129.0, 127.4–127.2, 123.8–123.2 (groups of phenanthroline carbons), 57.7–56.7 (group of pyrrolidine ring CH), 50.2–47.1, 27.1–26.3, 24.3–24.0, 21.0–20.8 (groups of pyrrolidine ring CH_2_); IR (ν, cm^−1^) 2986, 2898 (C–H stretching vibrations), 1646 (C=O), 1583, 1552, 1502, 1446 (C=C, C=N); HRMS (ESI–TOF) (*m*/*z*) [M + H]^+^ calcd for C_24_H_21_F_6_N_4_O_2_ 511.1563, found 511.1558.

*(4,7-dichloro-9-((2-(trifluoromethyl)pyrrolidin-1-yl)carbonyl)-1,10-phenanthrolin-2-yl)(2-(trifluoromethyl)pyrrolidin-1-yl)methanone* (**2**). Yield = 67% (1.94 g), white powder, m.p. 239–248 °C, R_f_ 0.39 (hexane:acetone 7:3); ^1^H NMR (400 MHz, CDCl_3_) δ 8.54–8.20 (m, 4H, phenanthroline protons), 7.13–5.08 (group of m, 2H, pyrrolidine ring CH), 4.82–2.96 (group of m, 4H, pyrrolidine ring CH_2_), 2.70–1.88 (group of m, 8H, pyrrolidine ring CH_2_); ^13^C NMR (101 MHz, CDCl_3_) δ 167.1–164.1 (group of CO), 153.1–152.2, 144.9–143.2, 127.5–127.3, 124.6–123.9 (groups of phenanthroline carbons), 57.7–56.8 (group of pyrrolidine ring CH), 50.2–47.3, 27.1–26.4, 24.3–23.9, 21.0–20.7 (groups of pyrrolidine ring CH_2_); IR (ν, cm^−1^) 2989, 2893 (C–H stretching vibrations), 1651, 1608 (C=O), 1572, 1533, 1445, 1408 (C=C, C=N); HRMS (ESI–TOF) (*m/z*) [M + H]^+^ calcd for C_24_H_19_Cl_2_F_6_N_4_O_2_ 579.0784, found 579.0791.

#### 3.3.2. Synthesis of Complexes **1**·Ln(NO_3_)_3_

A solution of lanthanoid nitrate (0.1 mmol) in dry acetonitrile (1 mL) was added dropwise to a solution of **L** (0.1 mmol) in chloroform (1 mL). After that the reaction mixture was concentrated in vacuo (~20 Torr) to 1/5 of the initial volume and treated with diethyl ether (2 mL). The resulting precipitate of the complex was filtered, washed with a fresh portion of ether, dried in air, then at 80 °C at ~2 Torr.

*(2-(trifluoromethyl)pyrrolidin-1-yl)(9-((2-(trifluoromethyl)pyrrolidin-1-yl)carbonyl)-1,10-phenanthrolin-2-yl)methanone lanthanum trinitrate* (**1·La(NO_3_)_3_**). Yield = 53 mg (63%). Yellowish powder. T_decomp_. 231 °C. ^1^H NMR (CD_3_CN): δ 8.83 (dd, 2H, Phen), 8.42 (dd, 2H, Phen), 8.23 (d, 2H, Phen H^5,6^), 5.45–5.24 (m, 2H, Pyr–CH), 4.18–3.88 (m, 5H, Pyr), 2.44–2.28 (m, 2H, Pyr), 2.19–2.01 (m, 5H, Pyr); ^19^F NMR (CD_3_CN): δ–74.16 (d, J = 7.5 Hz), –74.28 (d, J = 7.7 Hz); FT–IR (KBr, ν, cm^−1^): 1604 (C=O),1440, 1279 (ONO_2_); HRMS (ESI–TOF) (*m/z*) [**1****·La(NO_3_)_2_**]^+^ calcd for C_24_H_20_F_6_LaN_6_O_8_ 773.0305, found 773.0289.

*(2-(trifluoromethyl)pyrrolidin-1-yl)(9-((2-(trifluoromethyl)pyrrolidin-1-yl)carbonyl)-1,10-phenanthrolin-2-yl)methanone neodymium trinitrate* (**1·Nd(NO_3_)_3_**). Yield = 54 mg (64%). Yellow powder. T_decomp_. 235 °C. ^1^H NMR (CD_3_CN): δ 10.84 (ddd, 4H, Phen H^3,4,7,8^), 9.85 (d, 2H, Phen H^5,6^), 8.46 (d, 2H, N–CH), 4.86 (dt, 2H, N–CH_2_), 4.49 (s, 2H, N–CH_2_), 2.67 (s, 7H, CH_2_–CH_2_), 1.96 (s, 1H, CH_2_–CH_2_); ^13^C NMR (CD_3_CN): δ 179.4, 177.7, 156.5, 147.8, 146.5 (Phen), 146.3 (Phen), 139.6 (Phen), 139.4 (Phen), 133.5 (Phen C^5,6^), 133.1 (Phen C^5,6^), 62.7 (N–CH), 55.1 (N–CH_2_), 54.9 (N–CH_2_), 26.7 (CH_2_–CH_2_), 25.6 (CH_2_–CH_2_). ^19^F NMR (CD_3_CN): δ –72.66 (s, J = 7.5 Hz), –73.13 (s, J = 7.5 Hz); FT–IR (KBr, ν, cm^−1^): 1602 (C=O), 1463, 1293 (ONO_2_); HRMS (ESI–TOF) (*m/z*) [**1****·Nd(NO_3_)_2_**]^+^ calcd for C_24_H_20_F_6_NdN_6_O_8_ 776.0319, found 776.0318.

*(2-(trifluoromethyl)pyrrolidin-1-yl)(9-((2-(trifluoromethyl)pyrrolidin-1-yl)carbonyl)-1,10-phenanthrolin-2-yl)methanone europium trinitrate* (**1·Eu(NO_3_)_3_**). Yield = 74 mg (87%). Yellow powder. T_decomp_. 232 °C. ^1^H NMR (CD_3_CN): δ 6.54 (dd, 2H, Phen), 5.98 (d, 2H, Phen H^5,6^), 5.69 (d, 2H, Phen), 4.62 (t, 1H, Pyr), 4.32 (m, 2H, Pyr), 4.14–4.04 (m, 1H, Pyr), 3.34–3.06 (m, 2H, N–CH), 2.39–2.28 (m, 2H, Pyr), 2.17 (dd, 2H, Pyr), 2.03–1.967 (m, 4H, Pyr); ^13^C NMR (CD_3_CN): 149.6 (Phen), 149.4 (Phen), 127.1, 123.5, 123.2, 103.8 (CO), 89.7 (Phen), 89.3 (Phen), 58.7 (N–CH), 48.6 (N–CH_2_), 23.5, 23.2, 23.1; ^19^F NMR (CD_3_CN): δ –74.06 (d, J = 6.7 Hz), –74.50 (d, J = 6.8 Hz); FT–IR (KBr, ν, cm^−1^): 1601 (C=O), 1471, 1274 (ONO_2_); HRMS (ESI–TOF) (m/z) [**1****·Eu(NO_3_)_2_**]^+^ calcd for C_24_H_20_F_6_EuN_6_O_8_ 787.0454, found 787.0441.

*(2-(trifluoromethyl)pyrrolidin-1-yl)(9-((2-(trifluoromethyl)pyrrolidin-1-yl)carbonyl)-1,10-phenanthrolin-2-yl)methanone lutetium trinitrate* (**1·Lu(NO_3_)_3_**). Yield = 45 mg (52%). Yellow powder. T_decomp_. 258 °C. ^1^H NMR (CD_3_CN): δ 8.98 (d, J = 8.6 Hz, 2H, Phen), 8.66 (dd, J = 8.6, 2.5 Hz, 2H, Phen), 8.33 (d, J = 1.1 Hz, 2H, Phen H^5,6^), 5.37 (td, J = 7.6, 3.4 Hz, 2H, Pyr CH), 4.32 (q, J = 11.6, 8.5 Hz, 4H, Pyr N–CH_2_), 2.36–2.16 (8H, Pyr CH_2_–CH_2_); ^19^F NMR (CD_3_CN): δ –73.32 (d, J = 7.4 Hz), –73.46 (d, J = 7.5 Hz); FT–IR (KBr, ν, cm^−1^): 1597 (C=O), 1480, 1306 (ONO_2_); HRMS (ESI–TOF) (*m/z*) [**1****·Lu(NO_3_)_2_**]^+^ calcd for C_24_H_20_F_6_LuN_6_O_8_ 809.0649, found 809.0633.

For more spectral data, see Appendix A.

### 3.4. Quantum Chemistry Computations

Quantum chemistry computations were performed with the Gaussian 16, Revision C.01 program [40] using the density functional theory (PBE0) [42] and the def-2-TZVP basis set. Topological analysis of the ρ(**r**) function, calculations of the v(**r_bcp_**) and integration over interatomic zero-flux surfaces were performed using the AIMAll program. [43]

All expected critical points were found and the whole set of critical points in each system satisfies the Poincaré-Hopf rule.

Molecular geometries have been fully optimized (tolerance on gradient: 10^−7^ au) at PBE/L1 [44] level of theory using a PRIRODA-19 program developed by Laikov [45]. All stationary points on the potential energy surface (PES) were checked by vibrational analysis and none of them had imaginary frequencies.

Geometries of both ligands were optimized at B3LYP/6-31G(d,p) theoretical level, after that ESP maps were calculating using Gaussian 16 program [40].

## 4. Conclusions

To summarize, first trifluorinated phenanthrolinediamides **1** and **2** were synthesized using 2-CF_3_-pyrrolidine. Both new ligands are formed as a mixture of diastereomers–*meso*-(*R,S*) and *rac*-(*R,R* and *S,S*) in a 2:3 ratio. The structure of racemic **1** was determined by X-ray. A detailed study of the ^1^H, ^13^C and ^19^F NMR spectra as well as EXSY and ROESY techniques in solvents of different polarities and at different temperatures were performed for both ligands. It was found that each of the diastereomers in solutions exists in the form of three rotamers caused by hindered rotation along the amide bonds. The ratio of rotamers in equilibrium mixtures is determined by the polarity of the solvents and the temperature. The structures of diastereomers and rotamers for ligand **1** were calculated by the density functional theory. It was shown that both diamides **1** and **2** are efficient ligands to form complexes with lanthanoids nitrates (La, Nd, Eu, Lu) in acetonitrile media. The stability constants of the complexes were determined by spectrophotometric titration. It was found that ligand **1** can form complexes of 1:1 and 2:1 composition, whereas less basic ligand **2** forms only 1:1 complexes.

## Figures and Tables

**Figure 1 molecules-27-03114-f001:**
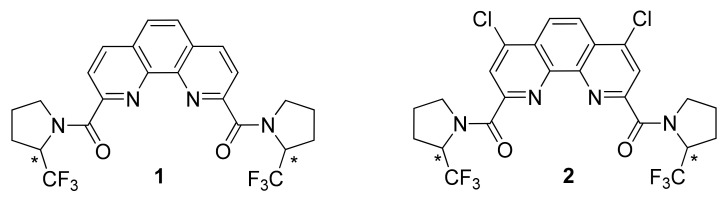
Trifluoromethylated phenanthrolinediamides **1** and **2**. Chiral centers marked with (*).

**Figure 2 molecules-27-03114-f002:**
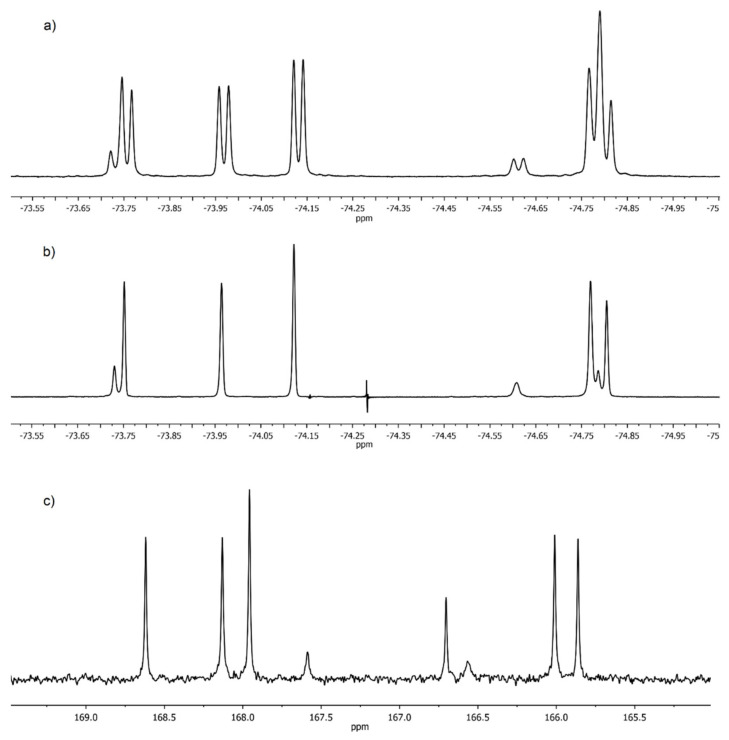
(**a**) ^19^F NMR spectrum; (**b**) ^19^F–{^1^H}–NMR spectrum; (**c**) Carbonyl part of ^13^C NMR of **1** in CDCl_3_ at 23 °C.

**Figure 3 molecules-27-03114-f003:**
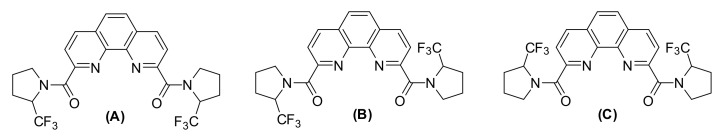
Structure of rotamers (**A**–**C**).

**Figure 4 molecules-27-03114-f004:**
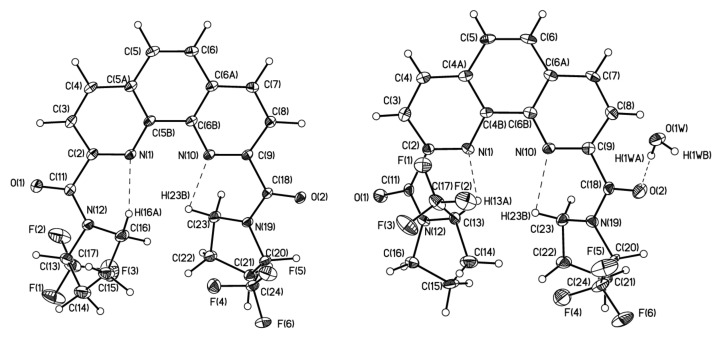
The general view of (left) **1A** and (right) **1B**·H_2_O in representation of atoms by thermal ellipsoids (*p* = 50%).

**Figure 5 molecules-27-03114-f005:**
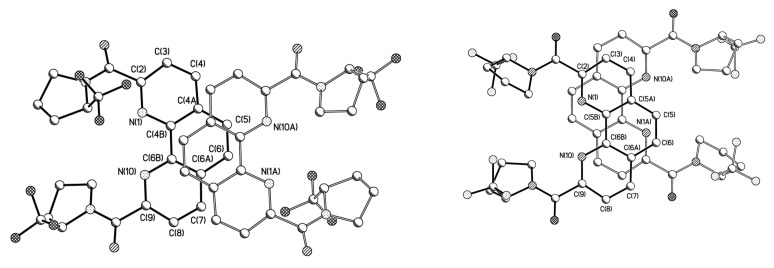
The molecular packing of (left) **1A** and (right) **1B**·H_2_O.

**Figure 6 molecules-27-03114-f006:**
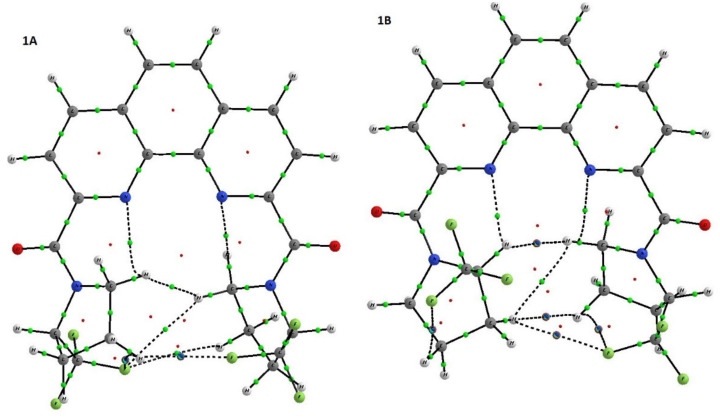
The molecular graph of two conformers of **1**. The CP (3,–1) are shown by green spheres.

**Figure 7 molecules-27-03114-f007:**
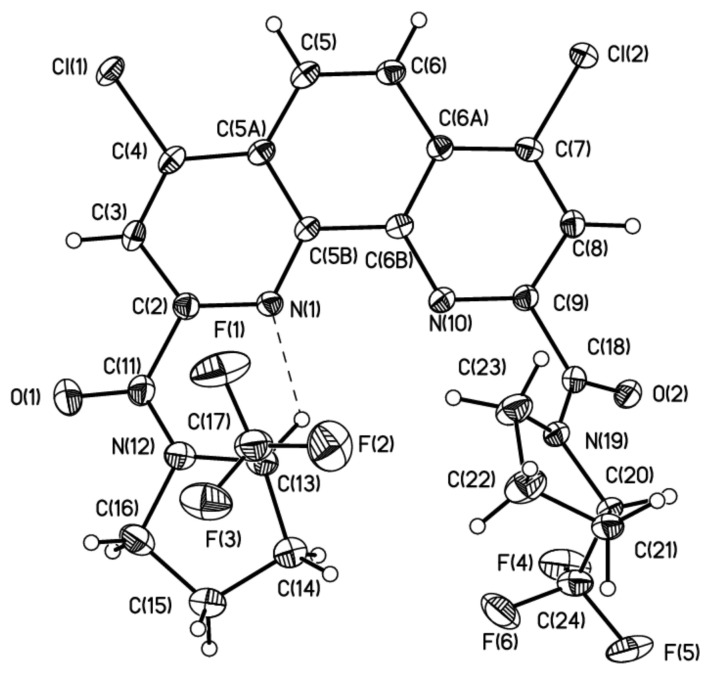
The general view of **2B** in representation of atoms by thermal ellipsoids (*p* = 50%).

**Figure 8 molecules-27-03114-f008:**
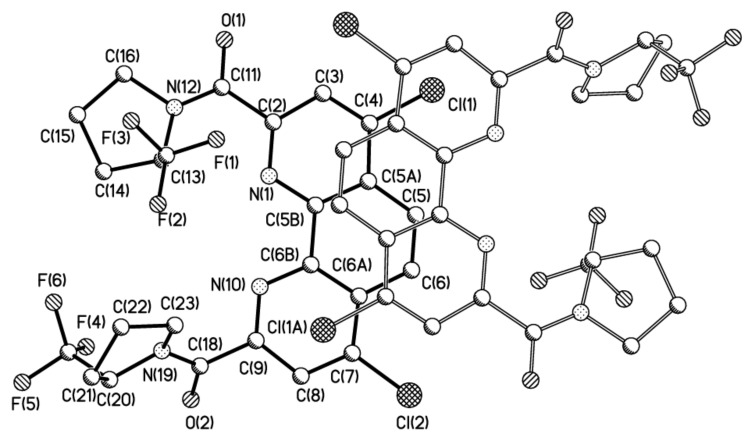
The molecular packing of **2B**.

**Figure 9 molecules-27-03114-f009:**
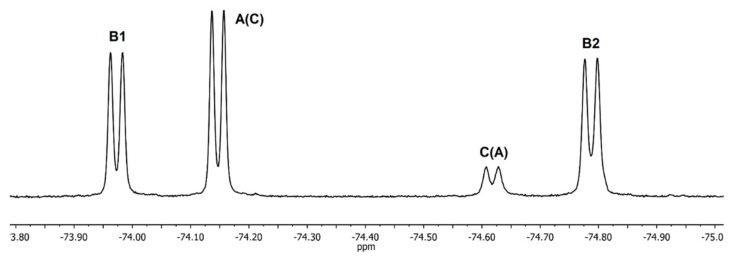
^19^F NMR spectrum of racemic **1** in CDCl_3_ at 23 °C.

**Figure 10 molecules-27-03114-f010:**
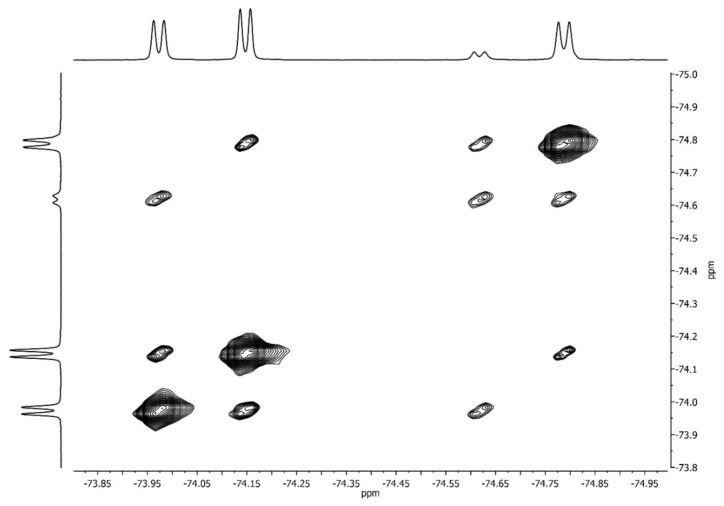
^19^F EXSY spectrum of racemic **1** in CDCl_3_ at 23 °C.

**Figure 11 molecules-27-03114-f011:**
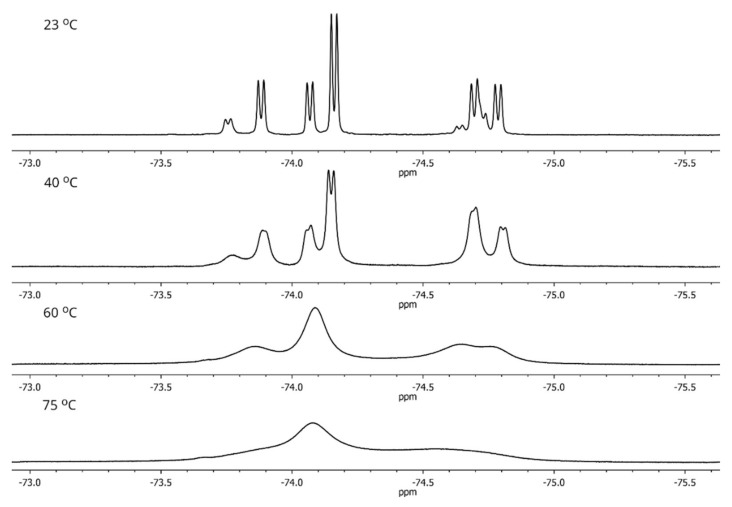
Temperature dependence of ^19^F NMR spectra of two diastereomers of **1** in toluene-d_8_.

**Figure 12 molecules-27-03114-f012:**
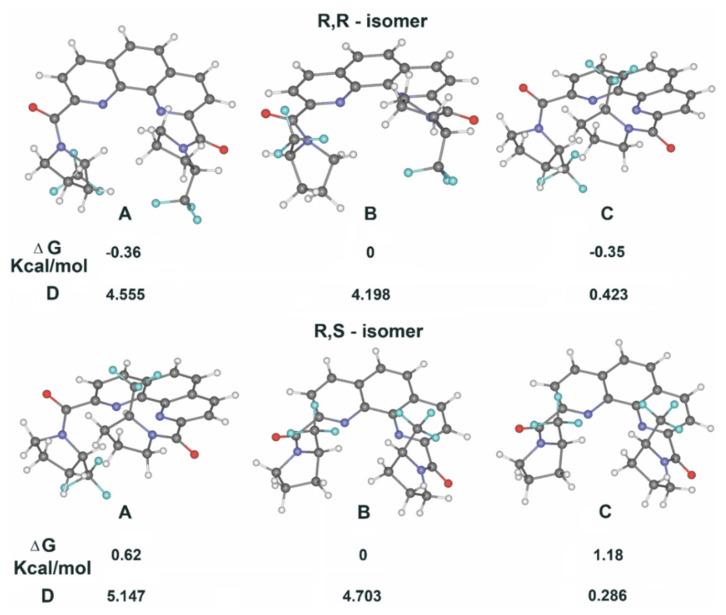
Calculated structures, energies and dipole moments in gas phase for rotamers of **1**.

**Figure 13 molecules-27-03114-f013:**
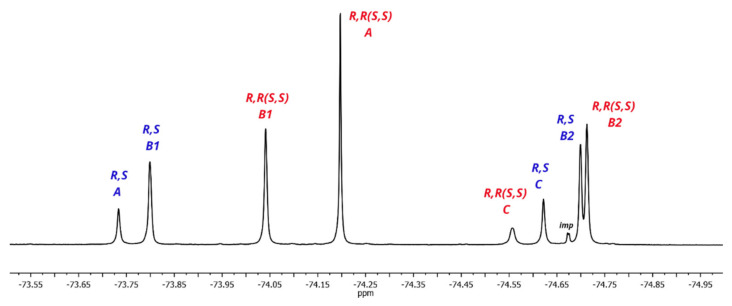
^19^F–{^1^H}–NMR of **2** in CDCl_3_ at 23 °C.

**Figure 14 molecules-27-03114-f014:**
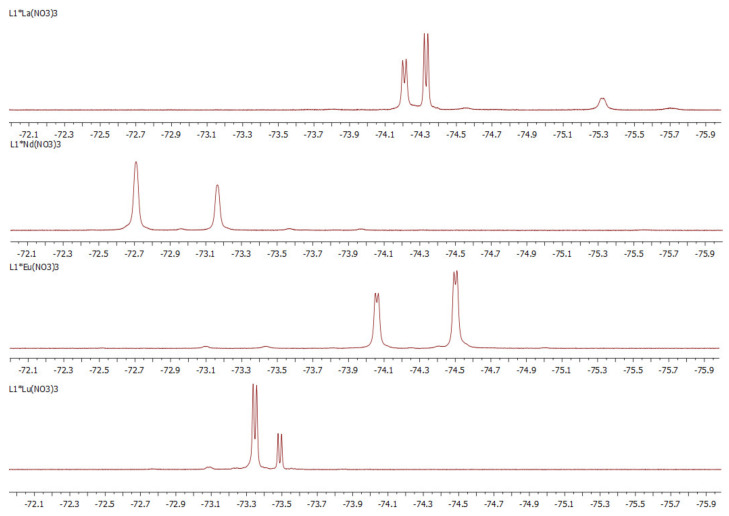
^19^F NMR spectra of **1·**Ln(NO_3_)_3_ complexes in CD_3_CN at 23 °C.

**Figure 15 molecules-27-03114-f015:**
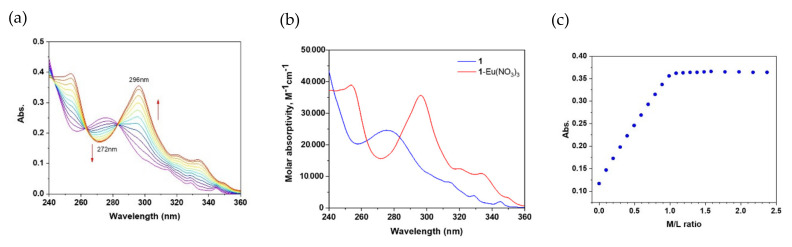
Spectrophotometric titration of 1 (ca. 10^−5^ mol/L) with Eu^3+^ ions (ca. 5 × 10^−4^ mol/L) in CH_3_CN solution (T = 25.0 ± 0.1 °C, I = 0 M, V_0_ = 2.0 mL): (**a**) absorption spectra; (**b**) molar absorptivities of free ligand 1 and Eu(III) complex calculated from spectral deconvolution; (**c**) titration curve at 296 nm (maximum absorption).

**Figure 16 molecules-27-03114-f016:**
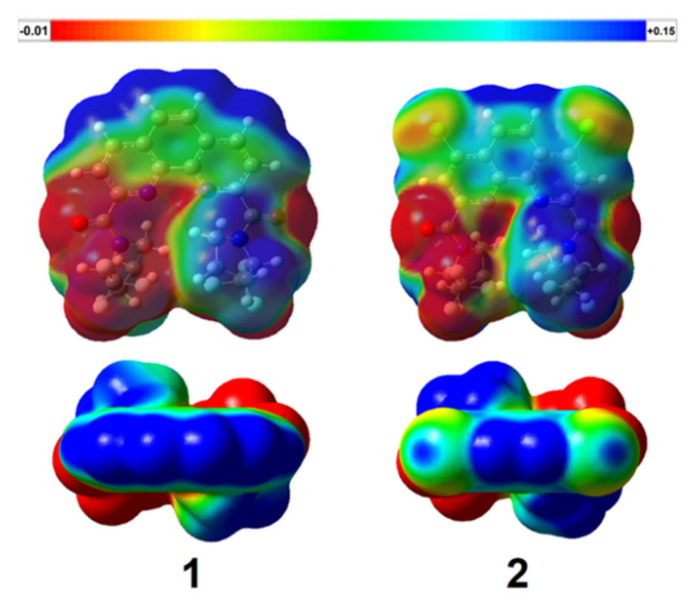
ESP maps of 1 and 2.

**Table 1 molecules-27-03114-t001:** Structural parameters of ligands **1** and **2** according to XRD and DFT data.

	XRD			DFT		
Bond Length (Å), Angles (°), Configuration	1A	1B·H_2_O	2B	1A	1B	2B
C(2)–C(11)	1.503(3)	1.502(6)	1.519(4)	1.509	1.51	1.514
C(11)–N(12)	1.356(3)	1.355(5)	1.346(4)	1.365	1.373	1.363
N(1)C(2)C(11)N(12)	26.8	37.3	25	32.17	20.928	29.429
C(13)N(12)C(11)O(1)	−2.7	−174.3	−178.7	−175.92	4.1	5.832
N(1)C(2)C(11)O(1)	−148.9	−140	−152.8	−146.1	−156.8	−147.751
C(9)–C(18)	1.511(3)	1.498(5)	1.502(4)	1.508	1.508	1.51126
C(18)–N(19)	1.365(2)	1.362(5)	1.355(4)	1.375	1.374	1.36977
N(10)C(9)C(18)N(19)	26.9	31.7	−47.9	26.06	28.8	−40.596
C(20)N(19)C(18)O(2)	1.3	−8.1	7.5	−4.35	−1.292	4.744
N(10)C(9)C(18)O(2)	−150.3	−148.6	131.6	−153.2	−148.738	−40.596
N(1)…H(13A)	2.46	2.16	2.13	2.168	2.38975	2.17
C(13)H(13A)N(1)	104.7	120.1	126.1	121.6	122.3	123.6
N(10)…H(23A)	2.40	2.43	2.56	2.464	2.44	2.55
C(23)H(23a)N(10)	102.6	98.3	105.7	97.9	99.6	101.1
C13	R*	R	R	R	R	R
C20	R	R	S *	R	R	S

* Configuration of the atom according to Cahn–Ingold–Prelog priority rules.

**Table 2 molecules-27-03114-t002:** Content of rotamers (%) of racemic **1** at different temperatures.

T/°C	A	B	C
21.5	33.0	59.0	8.0
30	33.1	58.0	8.9
40	33.2	57.3	9.5

**Table 3 molecules-27-03114-t003:** Solvent dependence of rotamers content for both diastereomers of **1** at 21.5 °C.

Solvent	ε	A		B		C	
		*racemic*	*meso*	*racemic*	*meso*	*racemic*	*meso*
toluene-d_8_	2.4	47.6	14.9	47.2	67.3	5.2	17.8
CDCl_3_	4.8	36.3	14.0	55.0	72	9.0	14.0
acetone-d_6_	20.7	40	35.0	53.5	54.0	6.5	11.0
C_6_D_5_NO_3_	34.8	85.5	48.0	14.5	52.0	0	0

**Table 4 molecules-27-03114-t004:** Ratio of rotamers **A–C** (%) for mixture of diastereomers of **2** in CDCl_3_ at 23 °C.

Diastereomer	A	B	C
Racemate	20.7	34.5	4.7
Meso	5.6	27.4	7.1

**Table 5 molecules-27-03114-t005:** Some characteristics of **1**·Ln(NO_3_)_3_ complexes.

	1	1·La(NO_3_)_3_	1·Nd(NO_3_)_3_	1·Eu(NO_3_)_3_	1·Lu(NO_3_)_3_
T_decomp_, °C	190–195	231	235	232	258
ν_C=O_	1646	1604	1602	1601	1597
∆ν_C=O_	-	42	44	45	49

**Table 6 molecules-27-03114-t006:** Stability constants (log β) of complexes of **1** and **2** with nitrates of La, Nd, Eu and Lu.

Ligand	Stability Constant	La^3+^	Nd^3+^	Eu^3+^	Lu^3+^
**1**	log β_1_	6.56 ± 0.02	6.66 ± 0.03	7.08 ± 0.01	7.21 ± 0.01
	log β_2_	12.19 ± 0.04	12.53 ± 0.07	N/A	N/A
**2**	log β_1_	5.57 ± 0.01	5.80 ± 0.01	6.10 ± 0.01	6.20 ± 0.02
	log β_2_	N/A	N/A	N/A	N/A

**Table 7 molecules-27-03114-t007:** Merz-Kollman charges of N_Phen_ and O_amide_ of 1 and 2.

Ligand	N_Phen_	O_amide_
**1**	−0.335/−0.446	−0.480/−0.486
**2**	−0.315/−0.421	−0.471/−0.480

## Data Availability

Not applicable.

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
