# Peer review of "First Trifluoromethylated Phenanthrolinediamides: Synthesis, Structure, Stereodynamics and Complexation with Ln(III)"

_molecules, 2022, doi:10.3390/molecules27103114_

Round 1
Reviewer 1 Report
In this manuscript, the authors reported 1,10-phenantrholine-2,9-diamides bearing СF3-groups and their complexes with Ln ions. The 1,10-phenantrholine-2,9-diamides were well characterized. However, for the Ln-based complexes, the crystal structures were not solved. The present characteristics are not enough. The coordination modes of the ligands are not determined.
Author Response
|
Thank you for your valuable comment. Indeed, an unambiguous determination of the coordination modes of the ligand in the complexes seems very desirable. To date, a large number of complexes of 1,10-phenanthroline-2,9-dicarboxylic acid diamides with a wide variety of substituents in the amide fragment have been described and characterized by X-ray analysis (see, for example, Lemport P.S. et al. Significant impact of lanthanide contraction on the structure of the phenanthroline complexes. Mendeleev Commun. 2021, 31, pp. 853–855. DOI: 10.1016/j.mencom.2021.11.028 and references therein). In all known cases, these ligands act as tetradentate. One of characteristic feature of such coordination is the shift of the band of carbonyl groups in the IR spectra. We also observe this fact for the complexes which are discussed in the present work. On the other hand, our current study aimed at finding out and elucidate the molecular dynamics for ligands in which both optically active centers are present and amide rotation of bulk fragments of molecules is possible. We are going to use the results of the current research on the study of even more complex molecules of this type. The complexes were obtained in order to confirm the principal possibility of their existence, as well as the effect of complexation on the molecular dynamics of ligands.
|
Reviewer 2 Report
Please see attached pdf file.

Author Response
|
The title suggests that the new ligands are useful for the separation of lanthanides and/or actinides, but the authors do not describe any experimental evidence for this. This assumed property may indeed be true - as related compounds show – but it is misleading to use it in this form, especially in the title. |
Due to the fact that the article is primarily devoted to the structural features of the obtained ligands, we propose the following title for our manuscript: «First Trifluoromethylated Phenanthrolinediamides: Synthesis, Structure, Stereodynamics and Complexation with Ln(III)» |
|
The sole contribution of the author A.S.A. seems to be funding acquisition – according to the ethical guidelines of MDPI this does not justify an authorship, but the name should be moved to the acknowledgement section |
Since this work is a multidisciplinary study, the number of authors is expected to be large. For A.S.A. we add “formal analysis” as a contribution role in “Author Contributions” |
|
The parent compound without the CF3-groups has been synthesized by a virtually identical route and has been described previously (J. Heterocyclic Chem., 41, (2004) 795-798) – but the work is neither mentioned nor cited – this is not acceptable. |
Now the parent compound is mentioned in lines 76 and 77 with corresponding refs: Synthesis of the “parent” pyrrolidine-dervied ligand was first published in 2004 [28]. Later, some properties of this compound were revealed [29]. |
|
line 30: a reference for the cited pKa has to be provided. |
Reference [1] was added. The text was corrected as follows: “These ligands have large size of the coordination cavity and relatively low Brönsted basicity [1].” |
|
line 166: In Figure 6 the two structures of the conformers of 1 have to be labelled unambiguously. |
Fixed. The conformers of 1 in Figure 6 have labelled and the pictures are swapped. |
|
line 180: The sentence refers to Fig. 9 not Fig. 6. |
Fixed. |
|
line 191: From the interpretation of the spectrum in Fig. 10, the experiment should rather be described as an EXSY, not as a NOESY experiment. |
Corrected everywhere in the text, marked with green |
|
line 210: Fig. 11 As the authors have recorded the temperature dependent 19F spectra, they should calculate the rotational barriers and compare them to the DFT derived values. |
The problem raised by the reviewer deserves more detailed consideration. A significant number of papers have been devoted to the dynamics of aryl carboxylic acid amides: 1. W. E. Stewart and T. H. Siddall, Chem. Rev., 1970, 70, 517–551, DOI: 10.1021/cr60267a001. 2. R. A. Bragg, J. Clayden, G. A. Morris and J. H. Pink, Chem.–Eur. J., 2002, 8, 1279–1289, DOI: 10.1002/1521-3765(20020315)8:6<1279::aid-chem1279>3.0.co;2-7. 3. E. A. Skorupska, R. B. Nazarski, M. Ciechaska, A. Jozwiak and A. KÅ‚ys, Tetrahedron, 2013, 69, 8147–8154, DOI: 10.1016/j.tet.2013.07.046. 4. S. Kim, J. Kim, J. Kim, D. Won, S.-K. Chang, W. Cha, K. Jeong, S. Ahn and K. Kwak, Molecules, 2018, 23(9), 2294, DOI:10.3390/molecules23092294 The barriers to rotation around the Ar–C(O) and R2N–C(O) bonds are quite close, and therefore these dynamic processes occur in a concert manner. As we have shown, these ligands coexist in the dynamic equilibrium of conformers, which further complicates the solution of the problem, since in addition to barriers, it is necessary to determine all the equilibrium constants between conformers when analyzing dynamic NMR spectra. As a result, the direct analysis of the spectra using a standard iterative procedure (complete line shape analysis) turned out to be impossible. At a qualitative level, it is possible to estimate the value of barriers to rotation by Phen-CO bonds by the half-widths of the spectrum lines in the slow exchange approximation. However, the actual accuracy (± 2-3 kcal/mol) is insufficient for comparison with the results of DFT calculations. More promising in such cases is the use of two-dimensional techniques, including the Accordion spectra (Yarnykh, V. L., Ustynyuk, Y. A., Lineshape Analysis of Two-Dimensional "Accordion" NMR Spectra for Quantitative Study of Multisite Chemical Exchange, J. Magn. Res. A, 1993, 102(2), 131-136; DOI: 10.1006/jmra.1993.1081). We planned to conduct such experiments |
|
line 303: bonging? Complexation of lanthanoids by 2? |
Term ‘complexation’ is now used instead |
|
line 320/321: the exact frequencies for 19F are missing. |
The frequency for 19F is 376.0 MHz. Corrected/added. |
|
line 321: the 1H/19F-HSQC spectra shown in the SI (Fig. 9,10,12) are not standard experiments (are they not rather HMBC experiments?) and should be described in more detail. |
Figure captions in SI have been corrected as such: Fig S9 and S10: FH_getCOR; Fig S11: HH gCosy; Fig S12: HC gHSQCAD Fig S13: HC gHMBCAD These experiments a standard pulse sequences included in Agilent 400-MR spectrometer software. |
|
the probeheads used for the NMR experiments are not reported, nor are the temperatures of the standard NMR experiments nor the concentration of the samples. |
Corrected/added. |
|
as two different NMR spectrometers are used (400 and 600 MHz), it is mandatory for each NMR figure (main text & SI), to state the Larmor frequency applied. |
Thank you for noticing this misspell. Only 400 MHz Agilent spectrometer was used in this work. All necessarily corrections were made in the text. |
|
line 314: a reference for the synthesis of 2-(Trifluoromethyl)pyrrolidine has to be provided. |
We added a phrase «Starting from (±)-2-(trifluoromethyl)pyrrolidine [30], we obtained new phenanthroline ligands 1 and 2.» to the main text (lines 75-76) and to the experimental part. The reference [30] is provided. |
|
In the experimental part a proper description of the synthesis should be provided – what is the “corresponding amine” if only racemic 2-(Trifluoromethyl)pyrrolidine was used. This holds also true for the “corresponding chloride” – a proper and unique chemical name should be used (IUPAC). |
Corrected, marked with green in the text. IUPAC names for the initial dichlorides were also added. |
|
According to the IUPAC recommendations the term lanthanide should not be used any more – use lanthanoid instead. |
Corrected everywhere in the text (with an exception for literary references) |
Reviewer 3 Report
This manuscript describes the synthesis and full characterization of two new CF3-containing 1,10-phenanthroline-2,9-diamide ligands. The authors have managed to elucidate the complicated solution behaviour using NMR and X-ray diffraction analysis and corroborated their results with theoretical studies. The coordination of these ligands to lanthanide ions across the series was investigated by NMR and IR techniques. Unfortunately, no crystals for X-ray studies were obtained. Interestingly, according to UV-vis spectrophotometric titrations, one ligand provided the possibility of forming 1:2 and 1:1 complexes with La and Nd, whereas the other ligand formed only 1:1 complexes.
Overall, this work presents an interesting study on new lanthanide coordinating ligands, which may serve in the future for the extraction/separation of f-elements.
Three points should be clarified before acceptance of publication:
a) the title says "Phenanthrolinediamides for Separation of f-Elements", however, there is not a single experiment in the manuscript concerning the separation of f-elements. Consequently, the titled should be changed.
b) recent work by Chengliang Xiao (e.g. Inorg Chem. 2021, 60, 8754) should be mentioned in the manuscript and the results should be compared to the current manuscript.
c) a characterization of the obtained lanthanide complexes by elemental analysis or by mass spectrometry would be desirable
Author Response
|
the title says "Phenanthrolinediamides for Separation of f-Elements", however, there is not a single experiment in the manuscript concerning the separation of f-elements. Consequently, the titled should be changed. |
we propose the following title for our manuscript: «First Trifluoromethylated Phenanthrolinediamides: Synthesis, Structure, Stereodynamics and Complexation with Ln(III)» |
|
recent work by Chengliang Xiao (e.g. Inorg Chem. 2021, 60, 8754) should be mentioned in the manuscript and the results should be compared to the current manuscript. |
Now this work is mentioned and cited as ref. [29] Recently we synthesized this substance and tried to work with it. However, it is not possible to adequately compare its extraction properties with our new trifluoromethylated ligands mainly because of pronounced amphiphilic properties of “parent” compound (LogP -0.08±1.30). Consequently, there are questions about the solubility of the resulting REE-complexes in an organic solvent/aqueous solution two-phase system and their distribution during liquid-liquid extraction tests. |
|
a characterization of the obtained lanthanide complexes by elemental analysis or by mass spectrometry would be desirable. |
HRMS-ESI data for all synthesized complexes was added to the experimental part |
Round 2
Reviewer 1 Report
The authors have addressed my concerns and the manuscript could be acceptable now.
Author Response
Thank you very much